# Incidence of COVID-19 reinfection among Midwestern healthcare employees

**Anne Rivelli**[1,2☉]*, **Veronica Fitzpatrick**[1,2☉], **Christopher Blair**[1,2‡], **Kenneth Copeland**[1,3‡], **Jon Richards**[1‡]

**1** Advocate Aurora Health, Downers Grove, IL, United States of America, **2** Advocate Aurora Research Institute, Downers Grove, IL, United States of America, **3** ACL Laboratories, Downers Grove, Illinois, United States of America

☉ These authors contributed equally to this work.
‡ CB, KC and JR also contributed equally to this work.
* Anne.rivelli@aah.org

**Data Availability Statement:** Data cannot be shared publicly without permission from Advocate Aurora Health. Data are available from the data analytics team, who acted as honest brokers on

## Abstract

Given the overwhelming worldwide rate of infection and the disappointing pace of vaccination, addressing reinfection is critical. Understanding reinfection, including longevity after natural infection, will allow us to better know the prospect of herd immunity, which hinges on the assumption that natural infection generates sufficient, protective immunity. The primary objective of this observational cohort study is to establish the incidence of reinfection of COVID-19 among healthcare employees who experienced a prior COVID-19 infection over a 10-month period. Of 2,625 participants who experienced at least one COVID-19 infection during the 10-month study period, 156 (5.94%) experienced reinfection and 540 (20.57%) experienced recurrence after prior infection. Median days were 126.50 (105.50–171.00) to reinfection and 31.50 (10.00–72.00) to recurrence. Incidence rate of COVID-19 reinfection was 0.35 cases per 1,000 person-days, with participants working in COVID-clinical and clinical units experiencing 3.77 and 3.57 times, respectively, greater risk of reinfection relative to those working in non-clinical units. Incidence rate of COVID-19 recurrence was 1.47 cases per 1,000 person-days. This study supports the consensus that COVID-19 reinfection, defined as subsequent infection ≥ 90 days after prior infection, is rare, even among a sample of healthcare workers with frequent exposure.

## Introduction

SARS-CoV-2, the virus that causes COVID-19, has been shroud in mystery since the first confirmed case was documented in Wuhan City, China in December 2019. A year and a half later, there have been over 190 million cases, 4 million deaths, and varying degrees of successful containment and mitigation [1]. The ultimate goal is global herd immunity for COVID-19, with the two main paths to achieving herd immunity being natural infection and vaccination [2]. After six months of mass vaccination efforts against SARS-CoV-2, preliminary data suggest extremely promising vaccine immunity results [3]. However, while some countries have vaccinated more than half of their populations, many lag behind [3].

this study, and are not authors on the publication. The research analyst meets the criteria for access to confidential data. Please contact ANDY MAREK - ANDREW.MAREK@AAH.ORG for more information.

**Funding:** This study was funded by Advocate Aurora Health (AAH). AAH had no role in study design, data collection and analysis, decision to publish, or preparation of the manuscript. The authors are all employees of AAH and receive salary from funding source. The authors, however, received no specific funding for this work.

**Competing interests:** The authors have declared that no competing interests exist.

Given the overwhelming worldwide rate of infection, especially with emerging variants, and the disappointing pace of vaccination, addressing reinfection is critical. Addressing reinfection, particularly the longevity of protection after natural infection, or natural immunity, will allow us to better understand the prospect of herd immunity, which hinges on the assumption that natural infection generates sufficient, protective immunity [2]. The primary aim of this paper is to provide longitudinal data on natural immunity after SARS-CoV-2 infection.

The incidence of true COVID-19 reinfection is challenging to document, as the extensive resources necessary to confirm reinfection have not been available or practical to employ clinically [4]. Confirmation of reinfection requires multiple polymerase chain reaction (PCR) tests, viral cultures, lab testing, and collection of clinical symptoms and epidemiological risk factors [4]. This has subsequently led to probable under-reporting of reinfection in scientific journals, as evidence based on these inaccessible resources have been required for formal reporting of COVID-19 reinfection [5]. Additionally, most individuals around the world who became infected during the first COVID-19 pandemic wave did not access a PCR or antibody test and/ or were not treated in the hospital, delaying efforts to recognize and track overall COVID-19 reinfection early on in the pandemic [5–9]. While the consensus is that reinfection is rare, more longitudinal studies focused on reinfection incidence in a variety of populations and time between confirmed infections will help corroborate this [3, 10–12].

The most up-to-date research suggests that infection provides natural immunity for at least three months [13] and immunity remains stable up to 6–8 months after the initial infection [12, 14]. Furthermore, the maximum duration of SARS-CoV-2 ribonucleic acid (RNA) shedding in the upper respiratory tract, indicating recurrence, has been reported to be between 83 and 104 days [15–18], meaning positive retesting after roughly 3 months of a prior positive PCR test, along with clinical criteria, favors confirmation of reinfection [19].

Based on the current available data, the Centers for Disease Control and Prevention (CDC) recently defined 90 days as the cut-off for retesting after a COVID-19 positive PCR test, given assumptions that primary infection can still result in a positive test for up to 90 days and that people with COVID-19 are protected from true reinfection for at least 90 days [20]. Additionally, one recent article proposed three detailed definitions of COVID-19 reinfection, specifically *confirmed reinfection* (characteristic clinical symptoms, positive PCR test result, positive viral culture if performed, >90 days from original infection, and viral RNA sequencing from both infections documenting unique strains); *clinical reinfection* (characteristic clinical symptoms, positive PCR test result, positive viral culture if performed, and epidemiological risk factor like known exposure with no other cause); and *epidemiological reinfection* (symptomatic or asymptomatic, positive PCR test result, positive viral culture if performed, and epidemiological risk factor like known exposure) [4]. Use of these definitions in research would promote more clarity and unity in results reporting.

This study aims to contributes longitudinal data on epidemiological reinfection in a large cohort of healthcare workers in the United States (US) with documented cases of COVID-19, as defined by positive PCR test results. This study is an extension of two previous studies among the same cohort that addressed factors related to seroprevalence of SARS-CoV-2 Immunoglobulin G (IgG) [21] and 3-month incidence of COVID-19 recurrence by SARS-CoV-2 IgG status [22]. In this prior publication, recurrence was used as an umbrella term that comprised numerous scenarios, including persistent illness, prolonged viral RNA shedding, increased virus replication, a different symptomatic viral infection in the presence of remnant SARS-CoV-2 RNA, and/or true reinfection with disease [4, 7, 9, 19]. This current study defines 'recurrence' the same, as all instances of subsequent reinfection after initial infection during the study period.

This study will address 10-month cumulative incidence of COVID-19 reinfection using the CDC's current guidelines. To provide context around reinfection, this study will also describe 10-month cumulative incidence of recurrence. We will also describe time to reinfection and recurrence, overall and stratified by clinical role in order to shine a light on the role of exposure frequency to SARS-CoV-2 in incidence and time to recurrence and reinfection.

## Materials and methods

This prospective cohort study recruited healthcare employees across a large Midwestern healthcare system, which consists of 26-hospitals and over 500 sites of care in Illinois and Wisconsin. SARS-CoV-2 IgG was measured in serum specimens obtained from all participants using the SARS-CoV-2 IgG Abbott Architect assay. Performance characteristics of the SARS-CoV-2 IgG assay were validated at ACL Laboratories, determining a sensitivity of 98.7% and specificity of 99.2% [23, 24]. To detect SARS-CoV-2, this study used the Aptima Panther SARS-CoV-2 Assay, which uses qualitative detection of RNA from SARS-CoV-2 isolated and purified nasopharyngeal, oropharyngeal and nasal swab specimens obtained from individuals who meet COVID-19 clinical and/or epidemiological criteria [25]. Both the SARS-CoV-2 Antibody Assay and the Aptima Panther TMA SARS-CoV-2 Assay were approved for use under Emergency Use Authorization in US laboratories certified under the Clinical Laboratory Improvement Amendments of 1988 [26]. Prior to recruitment, this study obtained approval by the Institutional Review Board (#20-168E).

### Participants

This study includes English- and Spanish-speaking adults ages $\geq$ 18 employed by the healthcare system as of June 8, 2020 (study initiation) who had at least one positive SARS-CoV-2 PCR test results in the system's Electronic Medical Record (EMR) system between March 1, 2020 and January 10, 2021. This sample of participants was drawn from the overarching study, which enrolled a convenience sample of 16,357 participants meeting the same inclusion criteria to test for SARS-CoV-2 IgG assay results between June 8, 2020 and July 10, 2020 [21]. After enrollment, all participants' positive SARS-CoV-2 PCR test results documented in the system's EMR between March 1, 2020 and January 10, 2021 were collected. It is implicit that team members were tested at a system-affiliated lab, if tested at all, due to no cost, convenience and employment implications.

### Procedures

On June 6, 2020, a detailed recruitment email was sent to all team members' work email addresses. The email provided instructions for participation in the study, including an alteration of consent and a study-specific passcode required for study registration. Interested team members were instructed to register in their active online health portal. Team members who met study inclusion criteria and completed a lab blood draw to test for SARS-CoV-2 IgG were participants in this study.

### Variables

Data gathered for this study included demographics and all system EMR-documented positive SARS-CoV-2 PCR test results for COVID-19 infection between March 1, 2020 and January 10, 2021, including days between study initiation and each positive SARS-CoV-2 PCR test result. Age was collected as continuous and collapsed into standard reporting categories (ages 18–24; 25–34; 35–44; 45–54; 55–64; 65+). Race/ethnicity included Hispanic; White, Non-Hispanic;

Black, Non-Hispanic; Asian, Non-Hispanic; American Indian, Non-Hispanic; or Mixed-race, Non-Hispanic (those who identified as two or more races). Sex included male and female. Clinical role category included COVID-clinical (participants working in a clinical capacity on COVID-19 designated units), clinical (participants working in a clinical capacity on a non-COVID-19 designated unit) or non-clinical (participants in non-clinical roles, both remote and on-site). Number of days between participants' study initiation and positive SARS-CoV-2 PCR test results were used to calculate person-time at risk and days to reinfection and recurrence.

The primary outcome, incidence of COVID-19 *reinfection*, represents the second documented SARS-CoV-2 positive PCR result for COVID-19 infection 90 or more days after a prior documented SARS-CoV-2 positive PCR result. For participants with more than two documented SARS-CoV-2 positive PCR results, the second documented infection that was closest to 90 or more days from the prior infection was included. For instance, one participant had seven total documented SARS-CoV-2 positive PCR results within the timeframe and their fifth documented infection was 92 days after their initial infection, so their initial and fifth infections and the days between were used in the reinfection analysis. This explains why there are more reinfection cases (156) than recurrence cases occurring at 90+ days (115). It should be noted that, if all first and last infections were included in reinfection analyses, there would be an additional 1162 person-days added to the overall person-time, reducing the incidence rate per 1,000 person-days a negligible amount. The secondary outcome, incidence of COVID-19 *recurrence*, represents the second documented SARS-CoV-2 positive PCR result after the initial documented SARS-CoV-2 positive PCR result, irrespective of time between positive results.

## Statistical methods

Data management and analysis were performed by the study research team and conducted using SAS statistical software (Version 9.4; SAS Institute, Cary, NC).

Descriptive statistics are reported as counts (%) or means (standard deviation) and median (interquartile range), as appropriate, particularly days to outcome. Demographic and baseline variables are also reported across primary and secondary outcome statuses. Corresponding measures of association include mean difference in age between those who did not experience reinfection or recurrence from those who did experience recurrence or reinfection and, for the remaining categorical variables, the odds ratio (OR), or the relative odds of participants of a given variable category experiencing COVID-19 recurrence or reinfection relative to the reference category of that variable. Variable reference levels were chosen based on lowest presumed risk. Corresponding p-values were generated from Student's T-tests for continuous variables and logistic regression Wald tests to represent differences in recurrence or reinfection.

Cumulative incidence of COVID-19 recurrence was calculated as number of participants who experienced a subsequent infection at/after 90 days of prior infection (reinfection) or who experienced a subsequent infection at all (recurrence) by total number of participants at risk of a subsequent infection between earliest positive PCR test result (March 1, 2020) and study end (January 10, 2021). Incidence rate (IR) was calculated as the number of participants at risk who experienced each outcome by person-days contributed to follow-up before the outcome was experienced or participant was censored at study end. The entire study period was counted as 315 days (the number of days between earliest positive PCR test result and study end). Incidence measures were calculated overall and by clinical role category. Incidence rate ratio (IRR) represents the relative IR between clinical role categories. Finally, categories of days to reinfection and recurrence are described as counts and percentages.

**Table 1. Incidence measures representing risk of COVID-19 reinfection.**

| REINFECTION | At Risk | Reinfection | Person Days | Cumulative Incidence | IR Per 1,000 Person-Days | IRR | 90–119 Days | 120–149 Days | 150–179 Days | 180+ Days |
|---|---|---|---|---|---|---|---|---|---|---|
| Overall | 2625 | 156 | 439974 | 5.94% | 0.354566 | - | 67 (42.95%) | 27 (17.31%) | 31 (19.87%) | 31 (19.87%) |
| Clinical Role | | | | | | | | | | |
| Non-Clinical | 231 | 4 | 38284 | 1.73% | 0.104482 | REF | 2 (50.00%) | 0 (0.00%) | 1 (25.00%) | 1 (25.00%) |
| Clinical | 1767 | 110 | 295172 | 6.23% | 0.372664 | 3.57 | 48 (43.64%) | 20 (18.18%) | 21 (19.09%) | 21 (19.09%) |
| COVID-Clinical | 627 | 42 | 106518 | 6.70% | 0.394300 | 3.77 | 17 (40.48%) | 7 (16.67%) | 8 (19.05%) | 10 (23.81%) |

## Results and discussion

### COVID-19 reinfection

Among all 2,625 total participants who experienced COVID-19 infection, defined by on positive SARS-CoV-2 PCR results, 156 (5.94%) experienced COVID-19 reinfection after the initial infection, contributing 439,974 total person-days of follow-up until they reached reinfection or study end (Table 1). Of these 156 participants who experienced reinfection, 42 (26.92%) had COVID-clinical roles, 110 (70.51%) had clinical roles, and 4 (2.56%) had non-clinical roles within the healthcare system. Cumulative incidence of reinfection within 10 months was 5.94% overall, 6.70% among COVID-clinical participants, 6.23% among clinical participants, and 1.73% among non-clinical participants. IRRs indicated 3.77 times and 3.57 times increased risk of COVID-19 reinfection among COVID-clinical and clinical participants, respectively, relative to non-clinical participants (Table 2).

### COVID-19 recurrence

Among all 2,625 total participants who experienced at least one COVID-19 infection, 540 (20.57%) experienced COVID-19 recurrence, contributing 368,085 total person-days of follow-up (Table 3). Of these 540 participants who experienced recurrence, 129 (23.89%) had COVID-clinical roles, 387 (71.67%) had clinical roles, and 24 (4.44%) had non-clinical roles within the healthcare system. Cumulative incidence of recurrence within 10 months was 20.57% overall, 20.57% among COVID-clinical participants, 21.90% among clinical participants, and 10.39% among non-clinical participants. IRRs indicated 2.07 times and 2.28 times increased risk of COVID-19 recurrence among COVID-clinical and clinical participants, respectively, relative to non-clinical participants (Table 4).

**Primary outcome: COVID-19 reinfection.** Among the 2,625 total participants who experienced COVID-19 infection, 156 (5.94%) experienced COVID-19 reinfection, or a subsequent positive SARS-CoV-2 test result 90 or more days later. Median time to recurrence was 126.50 (105.50, 171.00) days, with the majority of reinfection occurring between 90 and 119 days (42.95%). Participants working in COVID-clinical roles showed the greatest cumulative incidence of reinfection over 10 months (6.70%) followed closely by participants working in clinical roles (6.23%). Of those who experienced reinfection, almost all (97.40%) had COVID-clinical or clinical roles within the healthcare system, which put individuals in clinical roles at more than 3.5 times increased risk of COVID-19 reinfection as compared with individuals working remotely or in non-clinical roles.

**Secondary outcome: COVID-19 recurrence.** Among the 2,625 total participants who experienced COVID-19 infection, 540 (20.57%) experienced COVID-19 recurrence, or a subsequent positive SARS-CoV-2 PCR result. Median time to recurrence was 31.50 (10.00–72.00)

**Table 2. Demographics of Sample of Healthcare Employees, Overall and by COVID-19 Reinfection Status.**

| Variables of Interest | Overall Sample (N = 2625) | COVID-19 Infection (N = 2469; 94.06%) | COVID-19 Reinfection (N = 156; 5.94%) | Measures of Association^ (95% CI) | P-value |
|---|---|---|---|---|---|
| *Days to Reinfection* | - | - | 141.21 (42.80); 126.5 (105.5, 171.0) | - | - |
| 0–29 Days | - | - | 67 (42.95%) | - | - |
| 30–59 Days | - | - | 27 (17.31%) | - | |
| 60–89 Days | - | - | 31 (19.87%) | - | |
| 90+ Days | - | - | 31 (19.87%) | - | |
| *Age, mean (SD); median (IQR)* | 38.26 (11.62); 36 (29–47) | 38.29 (11.68); 35 (29–47) | 37.83 (10.64); 36.5 (29–46) | -0.46 (-2.34, 1.42) | 0.6313 |
| 18–24 | 200 (7.62%) | 184 (7.45%) | 16 (10.26%) | REF | 0.2031 |
| 25–34 | 1040 (39.62%) | 989 (40.06%) | 51 (32.69%) | 0.59 (0.33, 1.06) | |
| 35–44 | 634 (24.15%) | 587 (23.77%) | 47 (30.13%) | 0.92 (0.51, 1.66) | |
| 45–54 | 417 (15.89%) | 389 (15.76%) | 28 (17.95%) | 0.83 (0.44, 1.57) | |
| 55–64 | 306 (11.66%) | 292 (11.83%) | 14 (8.97%) | 0.55 (0.26, 1.16) | |
| 65+ | 28 (1.07%) | 28 (1.13%) | 0 (0.00%) | <0.001 (<0.001, >999.999) | |
| *Sex* | | | | | |
| Male | 361 (13.75%) | 347 (14.05%) | 14 (8.97%) | REF | 0.0769 |
| Female | 2264 (86.25%) | 2122 (85.95%) | 142 (91.03%) | 1.66 (0.95, 2.91) | |
| *Race*Ethnicity (N = 2,539)* | | N = 2390 | N = 149 | | |
| Non-Hispanic White | 1970 (77.59%) | 1853 (77.53%) | 117 (78.52%) | REF | 0.9891 |
| Non-Hispanic Black | 94 (3.70%) | 90 (3.77%) | 4 (2.68%) | 0.70 (0.25, 1.95) | |
| Non-Hispanic Asian | 181 (7.13%) | 171 (7.15%) | 10 (6.71%) | 0.93 (0.48, 1.80) | |
| Non-Hispanic American Indian | 3 (0.12%) | 3 (0.13%) | 0 (0.00%) | <0.001 (<0.001, >999.999) | |
| Non-Hispanic Mixed | 108 (4.25%) | 101 (4.23%) | 7 (4.70%) | 1.10 (0.50, 2.42) | |
| Hispanic | 183 (7.21%) | 172 (7.20%) | 11 (7.38%) | 1.01 (0.54, 1.92) | |
| *Clinical Role Category* | | | | | |
| Non-clinical | 231 (8.80%) | 227 (9.19%) | 4 (2.56%) | REF | 0.0284* |
| Clinical | 1767 (67.31%) | 1380 (66.19%) | 110 (70.51%) | 3.77 (1.38, 10.32)* | |
| COVID-clinical | 627 (23.89%) | 498 (23.88%) | 42 (26.92%) | 4.07 (1.44, 11.49)* | |

^Statistical significance indicated in this column represents Wald test p-values for direct differences between the variable level relative to the reference level of the same variable.

**Statistically significant at p<0.0001 for Wald tests if categorical or Student's T-test if continuous.

**Statistically significant at p<0.05 for Wald tests if categorical or Student's T-test if continuous.

**Table 3. Incidence measures representing risk of COVID-19 recurrence.**

| RECURRENCE | At Risk | Recurrence | Person-Days | Cumulative Incidence | IR Per 1,000 Person-Days | IRR | 0–29 Days | 30–59 Days | 60–89 Days | 90+ Days |
|---|---|---|---|---|---|---|---|---|---|---|
| Overall | 2625 | 540 | 368085 | 20.57% | 1.467052 | - | 257 (47.59%) | 116 (21.48%) | 52 (9.63%) | 115 (21.30%) |
| Clinical Role | | | | | | | | | | |
| Non-Clinical | 231 | 24 | 34490 | 10.39% | 0.695854 | REF | 16 (66.67%) | 2 (8.33%) | 2 (8.33%) | 4 (16.67%) |
| Clinical | 1767 | 387 | 244048 | 21.90% | 1.585754 | 2.28 | 173 (44.70%) | 88 (22.74%) | 42 (10.85%) | 84 (21.71%) |
| COVID-Clinical | 627 | 129 | 89547 | 20.57% | 1.440584 | 2.07 | 68 (52.71%) | 26 (20.16%) | 8 (6.20%) | 27 (20.93%) |

**Table 4. Demographics of sample of healthcare employees, overall and by COVID-19 recurrence status.**

| Variables of Interest | Overall Sample (N = 2625) | COVID-19 Infection (N = 2085; 79.43%) | COVID-19 Recurrence (N = 540; 20.57%) | Measures of Association^ (95% CI) | P-value |
|---|---|---|---|---|---|
| *Days to Recurrence* | - | - | 53.43 (57.88); 31.50 (10–72) | - | - |
| 0–29 Days | - | - | 257 (47.59%) | - | - |
| 30–59 Days | - | - | 116 (21.48%) | - | |
| 60–89 Days | - | - | 52 (9.63%) | - | |
| 90+ Days | - | - | 115 (21.30%) | - | |
| *Age, mean (SD); median (IQR)* | 38.26 (11.62); 36 (29–47) | 38.42 (11.78); 36 (29–47) | 37.64 (11.00); 35 (29–46) | -0.79 (-1.89, 0.31) | 0.1609 |
| 18–24 | 200 (7.62%) | 151 (7.24%) | 49 (9.07%) | REF | 0.1735 |
| 25–34 | 1040 (39.62%) | 832 (39.90%) | 208 (38.52%) | 0.77 (0.54, 1.10) | |
| 35–44 | 634 (24.15%) | 492 (23.60%) | 142 (26.30%) | 0.89 (0.61, 1.29) | |
| 45–54 | 417 (15.89%) | 332 (15.92%) | 85 (15.74%) | 0.79 (0.53, 1.18) | |
| 55–64 | 306 (11.66%) | 252 (12.09%) | 54 (10.00%) | 0.66 (0.43, 1.02) | |
| 65+ | 28 (1.07%) | 26 (1.25%) | 2 (0.37%) | 0.24 (0.05, 1.04) | |
| *Sex* | | | | | |
| Male | 361 (13.75%) | 295 (14.15%) | 66 (12.22%) | REF | 0.2471 |
| Female | 2264 (86.25%) | 1790 (85.85%) | 474 (87.78%) | 1.18 (0.89, 1.58) | |
| *Race*Ethnicity (N = 2,539)* | | | | | |
| Non-Hispanic White | 1970 (77.59%) | 1575 (77.93%) | 395 (76.25%) | REF | 0.7895 |
| Non-Hispanic Black | 94 (3.70%) | 76 (3.76%) | 18 (3.47%) | 0.94 (0.56, 1.60) | |
| Non-Hispanic Asian | 181 (7.13%) | 142 (7.03%) | 39 (7.53%) | 1.10 (0.76, 1.59) | |
| Non-Hispanic American Indian | 3 (0.12%) | 3 (0.15%) | 0 (0.00%) | <0.001 (<0.001, >999.999) | |
| Non-Hispanic Mixed | 108 (4.25%) | 80 (3.96%) | 28 (5.41%) | 1.40 (0.90, 2.18) | |
| Hispanic | 183 (7.21%) | 145 (7.17%) | 38 (7.34%) | 1.05 (0.72, 1.52) | |
| *Clinical Role Category* | | | | | |
| Non-clinical | 231 (8.80%) | 207 (9.93%) | 24 (4.44%) | REF | 0.0004* |
| Clinical | 1767 (67.31%) | 1380 (66.19%) | 387 (71.67%) | 2.42 (1.56, 3.75)* | |
| COVID-clinical | 627 (23.89%) | 498 (23.88%) | 129 (23.89%) | 2.23 (1.40, 3.56)* | |

^Statistical significance indicated in this column represents Wald test p-values for direct differences between the variable level relative to the reference level of the same variable.

**Statistically significant at p<0.0001 for Wald tests if categorical or Student's T-test if continuous.

**Statistically significant at p<0.05 for Wald tests if categorical or Student's T-test if continuous.

days. The majority of recurrence was documented within 60 days of the initial infection (68.07%), with most participants experiencing their second positive SARS-CoV-2 PCR test result within 30 days (47.59%). Participants working in clinical roles showed the greatest cumulative incidence of recurrence over 10 months (21.90%) followed closely by participants working in COVID-clinical roles (20.57%).

This study provides valuable data pertaining to the incidence and timing of COVID-19 reinfection and recurrence. Overall, this study corroborates previous studies that indicate reinfection is unlikely within a 10-month period, but not impossible. Both reinfection and recurrence were much more likely in clinical roles–in both COVID-clinical and non-COVID clinical units. Reinfection and recurrence, however, need to be addressed separately since recurrence alone, without the context of time, does not provide much information about the risk of true reinfection and natural immunity.

## Reinfection

Among the 2,625 total participants who experienced COVID-19 infection, 156 (5.94%) experienced COVID-19 reinfection, two positive tests at least 90 days apart, per the CDC definition, and within 10 months of that initial infection. The overall IR per 1,000 person-days was very low, indicating reinfection is rare. Interestingly, when comparing different clinical roles, the IRRs suggested 3.77 times and 3.57 times increased risk of COVID-19 reinfection among COVID-clinical and clinical participants, respectively, relative to non-clinical participants. This demonstrates that consistent re-exposure in a clinical setting may increase risk of reinfection. This study could validate previous speculation that reinfection is increased by continued exposure to SARS-CoV-2, even after a previous infection.

## Recurrence

Among the 2,625 total participants who experienced COVID-19 infection, 540 (20.57%) experienced COVID-19 recurrence, or at least two positive SARS-CoV-2 tests during this study period. Without viral testing, we don't know how many are accounted for by true reinfections or rather prolonged RNA shedding, persistent illness, or something else. Considering the difference in cumulative incidences between reinfection (5.94%) and recurrence (20.57%) shown in this study, it is likely that most recurrence in this study represents duplicate testing of the same infection, with the majority of recurrence occurring within 30 days (47.59%) and 60 days of the initial infection (69.07%). This, however, fails to explain why participants retested multiple times so close from the initial positive test. Return-to-work policies were based on resolution of COVID-19 symptoms and not retesting, even before the CDC released their 90-day retesting guidance. It is possible that healthcare workers had increased interest in their ongoing PCR test status and easier access to testing and, therefore, pursued retesting. As stated previously, this study did not assess symptomology or reasons for testing.

**Strengths.** This study enrolled and followed a large cohort of healthcare employees to determine risk of reinfection, as defined by the CDC, in a population likely to be re-exposed to COVID-19. This study provides much needed data to contribute to existing research on reinfection. PCR tests for COVID-19 infection were performed within system-affiliated labs, resulting in test performance and reporting consistency. All data was stored in EMR system and extracted by the healthcare system's analytics team, resulting in data collection consistency.

**Limitations.** There are several limitations to this study. Most important, there was no viral testing done to participants' blood samples, eliminating the ability to conclusively determine whether two SARS-CoV-2 test results in the same individual were due to true reinfection or recurrence. Second, extracted data for this study did not include symptomatology; therefore, we cannot determine 1) reasons participants tested multiple times, 2) sickness severity of participants with positive SARS-CoV-2 test results, or 3) commonalities among individuals with positive results. This information could have contributed to the body of literature that correlates viral load with the ability to transmit the virus [27]. Third, the majority of the cases of recurrence and reinfection occurred in participants who worked in clinical and COVID-clinical units which may limit the generalizability of the findings. Fourth, due to recruitment being conducted via work email, employees who are not frequent users of email, not comfortable using email, or could not navigate the instructions may have been unnecessarily excluded. Finally, because there is no universally accepted definition of *reinfection*, the study team used CDC retesting guidelines and some recently published guidance on proposed operational definitions of the terms to define reinfection and considered all subsequent positive test results to be recurrence.

## Discussion

Overall, this study indicates that reinfection is possible but unlikely, and both reinfection and recurrence are more likely among high-exposure groups like clinical healthcare workers. Of note, healthcare workers represented in this dataset are much more likely to be female, reflecting the fact that more women typically work in healthcare roles; however, as demonstrated in the tables, there was no statistically significant difference in reinfection or recurrence by sex. Given the increased odds of recurrence and reinfection noted in this study, special precautions and protections should prioritize clinical healthcare workers as they shoulder a large portion of disease, which is negatively affecting not only their physical health but mental health as well [28]. To start, individuals in high-exposure groups should continue to abide by previous public health precautions, irrespective of policy easement. Widespread vaccination and booster doses may be a solution to easing up on public health recommendations, but more long-term data is needed on vaccine efficacy, transmission and duration of protection in high exposure-risk populations. Vaccination rates will need to increase, as well, if we are ever to reach herd immunity since individuals will always be in higher-exposure groups. The current study end timeline was before the healthcare system began vaccinating front-line workers, which would have likely confounded the incidence of recurrence and reinfection. A future follow-up study using the same cohort will explore reinfection pre- and post-vaccination. Furthermore, post-COVID interventions are emerging [29–32], highlighting the need for research on the relationship between recurrence risk and post-COVID lifestyle changes, particularly behavioral modifications and post-COVID therapies.

## Acknowledgments

The authors would like to thank the AAH executive team, Public Affairs and Marketing, Andy Marek and Chris Blumberg in Research Analytics, ACL leadership and staff, the Health Informatics Technology (HIT) team, Advocate Aurora Research Institute (AARI), IRB, and Maureen Shields for early work on this project.

## Author Contributions

**Conceptualization:** Veronica Fitzpatrick, Christopher Blair, Kenneth Copeland, Jon Richards.

**Data curation:** Anne Rivelli, Veronica Fitzpatrick.

**Formal analysis:** Anne Rivelli.

**Investigation:** Veronica Fitzpatrick, Christopher Blair, Kenneth Copeland, Jon Richards.

**Methodology:** Anne Rivelli, Veronica Fitzpatrick, Kenneth Copeland, Jon Richards.

**Project administration:** Kenneth Copeland, Jon Richards.

**Resources:** Kenneth Copeland, Jon Richards.

**Supervision:** Veronica Fitzpatrick.

**Writing – original draft:** Anne Rivelli, Veronica Fitzpatrick.

**Writing – review & editing:** Anne Rivelli, Veronica Fitzpatrick, Christopher Blair, Kenneth Copeland, Jon Richards.

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
