## [Decision Letter · Decision Letter 0]

8 Sep 2021

PONE-D-21-26814Incidence of COVID-19 reinfection among Midwestern healthcare employeesPLOS ONE

Dear Dr. Rivelli,

Thank you for submitting your manuscript to PLOS ONE. After careful consideration, we feel that it has merit but does not fully meet PLOS ONE’s publication criteria as it currently stands. Therefore, we invite you to submit a revised version of the manuscript that addresses the points raised during the review process.

We look forward to receiving your revised manuscript.

Kind regards,

Alessandro de Sire, M.D.

Academic Editor

PLOS ONE

Journal Requirements:

"This study was funded internally. The funders had no role in study design, data collection and analysis, decision to publish, or preparation of the manuscript. The authors received no direct financial support for the research, authorship, and/or publication of this article."

"The authors would like to thank the many people at Advocate Aurora Health who supported this study,

specifically the executive team who financially supported staff testing, Public Affairs and Marketing, Andy

Marek and Chris Blumberg in Analytics, ACL leadership and staff, the Health Informatics Technology (HIT)

team, Advocate Aurora Research Institute (AARI), IRB, and Maureen Shields for early work on this project. "

"This study was funded internally. The funders had no role in study design, data collection and analysis, decision to publish, or preparation of the manuscript. The authors received no direct financial support for the research, authorship, and/or publication of this article."

6. We note you have included a table to which you do not refer in the text of your manuscript. Please ensure that you refer to Tables 1 to 4 in your text; if accepted, production will need this reference to link the reader to the Table.

Reviewers' comments:

Reviewer's Responses to Questions

**Comments to the Author**

1. Is the manuscript technically sound, and do the data support the conclusions?

Reviewer #1: Yes

Reviewer #2: Yes

2. Has the statistical analysis been performed appropriately and rigorously? 

Reviewer #1: Yes

Reviewer #2: Yes

3. Have the authors made all data underlying the findings in their manuscript fully available?

Reviewer #1: Yes

Reviewer #2: Yes

4. Is the manuscript presented in an intelligible fashion and written in standard English?

Reviewer #1: Yes

Reviewer #2: Yes

5. Review Comments to the Author

Reviewer #1: Dear Authors,

The manuscript is very interesting especially considering the hot topic in this specific historical moment. Moreover, the results are intriguing given the high number of subjects assessed. However, several critical issues should be addressed to improve the paper.

My major concern is that many factors that potentially might affect reinfection/recurrence of COVID 19 (such as pharmacological treatment of first-time infection, post-disease rehabilitation/interventions) have not been assessed. Therefore, the study results might be significantly affected by these confounders. Concurrently, patients excluded by the study analysis should be clarified in the Results section, characterizing at least the main cause of exclusions.

Furthermore, the Limitations subsection should underline that the specific population considered (the health care workers) severely limits the generalizability of the study results.

Lastly, the manuscript should be improved following the “ Submission Guidelines” of the Journal.

Major revisions

INTRODUCTION. This Section should be largely improved, underlining the health system burden due to medical care and disabling sequelae, mentioning the role of post-COVID therapies/behavioral modifications in the lifestyle change and risk of recurrence.

According to this, you should cite the following references:

• Curci C et al. Functional outcome after inpatient rehabilitation in post intensive care unit COVID-19 patients: findings and clinical implications from a real-practice retrospective study. Eur J Phys Rehabil Med. 2021;57(3):443-450. doi:10.23736/S1973-9087.20.06660-5

• Iddir M et al. Strengthening the Immune System and Reducing Inflammation and Oxidative Stress through Diet and Nutrition: Considerations during the COVID-19 Crisis. Nutrients. 2020;12(6):1562. Published 2020 May 27. doi:10.3390/nu12061562

• Ferraro F et al. COVID-19 related fatigue: Which role for rehabilitation in post-COVID-19 patients? A case series. J Med Virol. 2021;93(4):1896-1899. doi:10.1002/jmv.26717

• Andrenelli E et al. Systematic rapid living review on rehabilitation needs due to COVID-19: update to May 31st, 2020. Eur J Phys Rehabil Med. 2020;56(4):508-514. doi:10.23736/S1973-9087.20.06435-7

Moreover, anecdotical data should be avoided, while each sentence should be followed by a reference.

METHODS. Please data and numbers should be presented in the Results section rather than the Methods Section. Moreover, the examples of single cases enrolled in the study should be avoided. Please, refer to other studies to better clarify the study methods.

RESULTS. Patients excluded by the study analysis should be clarified in the results section, characterizing at least the main cause of exclusions. Moreover, the number of patients excluded should be discussed in the discussion section, better characterizing the risk of BIAS due to the enrollment methods.

RESULTS. Data about the viral load might provide useful information to compare first-time infections and recurrences. Please improve the limitation section accordingly if you could not provide these data.

CONCLUSION. This section should follow the “Discussion” Section according to the “Submission Guidelines” of the Journal.

DISCUSSION. Differences between female and male groups should be discussed in this section, providing some explanation to this interesting data.

DISCUSSION. Limitations. Page 13. The Limitation subsection should underline that the specific population considered (the health care workers) severely limits the generalizability of the study results.

DISCUSSION. Limitations. Page 13. The Limitation subsection should be improved highlighting that lack of data about pharmacological treatment/medical care of first-time infection might potentially affect the reinfection/recurrence.

Minor revisions

WHOLE TEXT. You should format your work according to the “Submission Guidelines” of the Journal. Therefore, you should include page numbers and line numbers in the manuscript file. Furthermore, the reference number in the text must be cited in square brackets.

WHOLE TEST. Please use the bold type, 18pt font for the Titles of the sections, following the Submission Guidelines” of the Journal.

ABSTRACT. You should improve this section following the “Submission Guidelines” of the Journal, Moreover, this section should not exceed 300 words.

METHODS. This section should be named “Material and Methods” accordingly with “Submission Guidelines” of the Journal

Reviewer #2: The article is of scientific interest and in line with the aims of the journal. The authors guidelines have been respected and the article does not require a revision of the English language.

There are minor concerns to be addressed.

ABSTRACT

It is well structured.

KEYWORDS

In order to increase the visibility of the article, do not use keywords already present in the title.

INTRODUCTION

The introduction fully discusses the topics covered. The purpose of the study is well specified.

MATERIALS AND METHODS

I suggest to give information about the number of subjects included just into the results section and not also in the methods.

Considering the current pandemic and its global duration, your choice of a convenience sample appears to be methodologically correct. Nevertheless, it could be useful to better clarify how the recruitment was conducted. Was it necessary to express informed consent and therefore choose whether to join the study or not? If yes, do You think there may be a significant number of healthcare workers who did not want to join the study despite having an already properly registered first infection?

RESULTS AND CONCLUSIONS

They are adequate.

DISCUSSION

A strength of this research seems to be overshadowed. It examines the incidence of COVID-19 reinfection and recurrence in a population of healthcare workers: what useful implications can be derived from it for the purpose of protecting these workers, especially from a psychological point of view, given their awareness of the risk of infection and reinfection? I suggest You to deep these aspects in the implications-section of the discussion using the following reference:

Farì G, de Sire A, Giorgio V, Rizzo L, Bruni A, Bianchi FP, Zonno A, Pierucci P, Ranieri M, Megna M. Impact of COVID-19 on the mental health in a cohort of Italian rehabilitation healthcare workers. J Med Virol. 2021 Aug 13. doi: 10.1002/jmv.27272. Epub ahead of print. PMID: 34387886.

TABLES

The tables are clear and adequately complement the text.

6. PLOS authors have the option to publish the peer review history of their article (what does this mean?). If published, this will include your full peer review and any attached files.

Reviewer #1: **Yes: **Lorenzo Lippi

Reviewer #2: No

---

## [Author Response · Author response to Decision Letter 0]

2 Nov 2021

REVIEWER 1

The manuscript is very interesting especially considering the hot topic in this specific historical moment. Moreover, the results are intriguing given the high number of subjects assessed. However, several critical issues should be addressed to improve the paper.

My major concern is that many factors that potentially might affect reinfection/recurrence of COVID 19 (such as pharmacological treatment of first-time infection, post-disease rehabilitation/interventions) have not been assessed. Therefore, the study results might be significantly affected by these confounders. Concurrently, patients excluded by the study analysis should be clarified in the Results section, characterizing at least the main cause of exclusions.

Furthermore, the Limitations subsection should underline that the specific population considered (the health care workers) severely limits the generalizability of the study results.

Lastly, the manuscript should be improved following the “ Submission Guidelines” of the Journal.

Thank you for this comment, this is an epidemiological look at the incidence of reinfection with no information on symptoms or treatments. We understand this is a limitation and it has been addressed in the limitations section. Patients were not “excluded” from this study for any reason other than they did not meet the criteria of having more than one positive covid test. We do not feel that this being a healthcare system setting is a limitation given that the original sample included remote employees. The fact that the clinical workers had higher exposure to COVID and did experience reinfection at higher rates is one of the primary findings. However, I have added a sentence about this limitation to make it more explicit. Updated submission via guidelines

INTRODUCTION. This Section should be largely improved, underlining the health system burden due to medical care and disabling sequelae, mentioning the role of post-COVID therapies/behavioral modifications in the lifestyle change and risk of recurrence.

According to this, you should cite the following references:

• Curci C et al. Functional outcome after inpatient rehabilitation in post intensive care unit COVID-19 patients: findings and clinical implications from a real-practice retrospective study. Eur J Phys Rehabil Med. 2021;57(3):443-450. doi:10.23736/S1973-9087.20.06660-5

• Iddir M et al. Strengthening the Immune System and Reducing Inflammation and Oxidative Stress through Diet and Nutrition: Considerations during the COVID-19 Crisis. Nutrients. 2020;12(6):1562. Published 2020 May 27. doi:10.3390/nu12061562

• Ferraro F et al. COVID-19 related fatigue: Which role for rehabilitation in post-COVID-19 patients? A case series. J Med Virol. 2021;93(4):1896-1899. doi:10.1002/jmv.26717

• Andrenelli E et al. Systematic rapid living review on rehabilitation needs due to COVID-19: update to May 31st, 2020. Eur J Phys Rehabil Med. 2020;56(4):508-514. doi:10.23736/S1973-9087.20.06435-7

Moreover, anecdotical data should be avoided, while each sentence should be followed by a reference.

Thank you for this suggestion. As stated above this is an epidemiological look at reinfection and not a paper on the burden of covid. Although the burden is very high, this paper is assessing frequent exposure with the odds of experiencing reinfection in a health system where people are likely to get tested for job related reasons and because of the convenience of testing.

Anecdotal data is not being used, it’s making the case for the importance of assessing reinfection. The manuscript was double checked and all declarative statements have a citation. 

METHODS. Please data and numbers should be presented in the Results section rather than the Methods Section. Moreover, the examples of single cases enrolled in the study should be avoided. Please, refer to other studies to better clarify the study methods.

There are no results presented in the methods section. There is an example given to help clarify the operational definitions of recurrence and reinfection.

RESULTS. Patients excluded by the study analysis should be clarified in the results section, characterizing at least the main cause of exclusions. Moreover, the number of patients excluded should be discussed in the discussion section, better characterizing the risk of BIAS due to the enrollment methods.

CONCLUSION. This section should follow the “Discussion” Section according to the “Submission Guidelines” of the Journal.

UPDATED

DISCUSSION. Differences between female and male groups should be discussed in this section, providing some explanation to this interesting data.

There was no statistical difference between males and females in whether they were more likely to experience recurrence or reinfection. The huge difference in percentage is due to the fact that women are much, much more likely to be in clinical roles in a healthcare setting – typically as nurses or medical assistants.

DISCUSSION. Limitations. Page 13. The Limitation subsection should underline that the specific population considered (the health care workers) severely limits the generalizability of the study results.

Added; lines: 257-259

DISCUSSION. Limitations. Page 13. The Limitation subsection should be improved highlighting that lack of data about pharmacological treatment/medical care of first-time infection might potentially affect the reinfection/recurrence.

Lines 253-256

WHOLE TEXT. You should format your work according to the “Submission Guidelines” of the Journal. Therefore, you should include page numbers and line numbers in the manuscript file. Furthermore, the reference number in the text must be cited in square brackets.

Updated - citations are in parentheses, as per guidelines

WHOLE TEST. Please use the bold type, 18pt font for the Titles of the sections, following the Submission Guidelines” of the Journal.

Updated

ABSTRACT. You should improve this section following the “Submission Guidelines” of the Journal, Moreover, this section should not exceed 300 words.

Updated

METHODS. This section should be named “Material and Methods” accordingly with “Submission Guidelines” of the Journal

Updated

REVIEWER 2

The article is of scientific interest and in line with the aims of the journal. The authors guidelines have been respected and the article does not require a revision of the English language.

ABSTRACT: It is well structured.

NO CHANGES

KEYWORDS: In order to increase the visibility of the article, do not use keywords already present in the title.

WILL UPDATE IN ONLINE PORTAL

INTRODUCTION: The introduction fully discusses the topics covered. The purpose of the study is well specified.

NO CHANGES

MATERIALS AND METHODS: I suggest to give information about the number of subjects included just into the results section and not also in the methods.

Number of subjects is in results and discussion section under ‘Covid-19 Reinfection’ header; lines: 180-182

Considering the current pandemic and its global duration, your choice of a convenience sample appears to be methodologically correct. Nevertheless, it could be useful to better clarify how the recruitment was conducted. Was it necessary to express informed consent and therefore choose whether to join the study or not? If yes, do You think there may be a significant number of healthcare workers who did not want to join the study despite having an already properly registered first infection?

Recruitment is addressed in Procedures; lines: 126-130

An alteration of consent was attached to the email and this is explained in Procedures; lines: 127-130

RESULTS AND CONCLUSIONS: They are adequate.

NO CHANGES

DISCUSSION

A strength of this research seems to be overshadowed. It examines the incidence of COVID-19 reinfection and recurrence in a population of healthcare workers: what useful implications can be derived from it for the purpose of protecting these workers, especially from a psychological point of view, given their awareness of the risk of infection and reinfection? I suggest You to deep these aspects in the implications-section of the discussion using the following reference:

Farì G, de Sire A, Giorgio V, Rizzo L, Bruni A, Bianchi FP, Zonno A, Pierucci P, Ranieri M, Megna M. Impact of COVID-19 on the mental health in a cohort of Italian rehabilitation healthcare workers. J Med Virol. 2021 Aug 13. doi: 10.1002/jmv.27272. Epub ahead of print. PMID: 34387886.

ADDED; Lines: 259-261

TABLES

The tables are clear and adequately complement the text.

NO CHANGES

---

## [Decision Letter · Decision Letter 1]

29 Nov 2021

PONE-D-21-26814R1Incidence of COVID-19 reinfection among Midwestern healthcare employeesPLOS ONE

Dear Dr. Rivelli,

Thank you for submitting your manuscript to PLOS ONE. After careful consideration, we feel that it has merit but does not fully meet PLOS ONE’s publication criteria as it currently stands. Therefore, we invite you to submit a revised version of the manuscript that addresses the points raised during the review process.

Dear Authors,

the paper could be considered suitable for publication after minor revisions.

Best regards

We look forward to receiving your revised manuscript.

Kind regards,

Alessandro de Sire, M.D.

Academic Editor

PLOS ONE

Journal Requirements:

Reviewers' comments:

Reviewer's Responses to Questions

**Comments to the Author**

1. If the authors have adequately addressed your comments raised in a previous round of review and you feel that this manuscript is now acceptable for publication, you may indicate that here to bypass the “Comments to the Author” section, enter your conflict of interest statement in the “Confidential to Editor” section, and submit your "Accept" recommendation.

Reviewer #1: (No Response)

Reviewer #2: All comments have been addressed

2. Is the manuscript technically sound, and do the data support the conclusions?

Reviewer #1: Yes

Reviewer #2: Yes

3. Has the statistical analysis been performed appropriately and rigorously? 

Reviewer #1: Yes

Reviewer #2: Yes

4. Have the authors made all data underlying the findings in their manuscript fully available?

Reviewer #1: Yes

Reviewer #2: Yes

5. Is the manuscript presented in an intelligible fashion and written in standard English?

Reviewer #1: No

Reviewer #2: Yes

6. Review Comments to the Author

Reviewer #1: Dear Authors,

The manuscript is very interesting especially considering the hot topic in this specific historical moment. Moreover, the results are intriguing given the high number of subjects assessed. You have significantly improved the manuscript; however, a few issues should be addressed.

Major revisions

INTRODUCTION:

I understand the Author's point of view. However, I abide by my previous comment. The introduction should highlight the needing for evidence in this topic in this specific historical moment.

Therefore, it should be highlighted the health system burden due to medical care and disabling sequelae, mentioning the role of post-COVID therapies/behavioral modifications in the lifestyle change and risk of recurrence.

According to this, you should cite the following references:

• Curci C et al. Functional outcome after inpatient rehabilitation in post intensive care unit COVID-19 patients: findings and clinical implications from a real-practice retrospective study. Eur J Phys Rehabil Med. 2021;57(3):443-450. doi:10.23736/S1973-9087.20.06660-5

• Iddir M et al. Strengthening the Immune System and Reducing Inflammation and Oxidative Stress through Diet and Nutrition: Considerations during the COVID-19 Crisis. Nutrients. 2020;12(6):1562. Published 2020 May 27. doi:10.3390/nu12061562

• Ferraro F et al. COVID-19 related fatigue: Which role for rehabilitation in post-COVID-19 patients? A case series. J Med Virol. 2021;93(4):1896-1899. doi:10.1002/jmv.26717

• Andrenelli E et al. Systematic rapid living review on rehabilitation needs due to COVID-19: update to May 31st, 2020. Eur J Phys Rehabil Med. 2020;56(4):508-514. doi:10.23736/S1973-9087.20.06435-7

METHODS. The number of patients enrolled in a prospective study represents a study result. Please present these data in the Results section rather than the Methods Section.

DISCUSSION. Please discuss the huge difference in percentage between females and males probably because women are much, much more likely to be in clinical roles in a healthcare setting – typically as nurses or medical assistants.

RESULTS AND DISCUSSION. This section should be divided into two different sections. The manuscript form is not suitable for the journal without the suggested improvements.

Reviewer #2: The final version of the paper seems well structured. All the required corrections have been done. In my opinion, the study is now acceptable for publication.

7. PLOS authors have the option to publish the peer review history of their article (what does this mean?). If published, this will include your full peer review and any attached files.

Reviewer #1: No

Reviewer #2: No

---

## [Author Response · Author response to Decision Letter 1]

13 Dec 2021

Below are the points raised and subsequently addressed:

INTRODUCTION:

I understand the Author's point of view. However, I abide by my previous comment. The introduction should highlight the needing for evidence in this topic in this specific historical moment.

Therefore, it should be highlighted the health system burden due to medical care and disabling sequelae, mentioning the role of post-COVID therapies/behavioral modifications in the lifestyle change and risk of recurrence.

According to this, you should cite the following references:

• Curci C et al. Functional outcome after inpatient rehabilitation in post intensive care unit COVID-19 patients: findings and clinical implications from a real-practice retrospective study. Eur J Phys Rehabil Med. 2021;57(3):443-450. doi:10.23736/S1973-9087.20.06660-5

• Iddir M et al. Strengthening the Immune System and Reducing Inflammation and Oxidative Stress through Diet and Nutrition: Considerations during the COVID-19 Crisis. Nutrients. 2020;12(6):1562. Published 2020 May 27. doi:10.3390/nu12061562

• Ferraro F et al. COVID-19 related fatigue: Which role for rehabilitation in post-COVID-19 patients? A case series. J Med Virol. 2021;93(4):1896-1899. doi:10.1002/jmv.26717

• Andrenelli E et al. Systematic rapid living review on rehabilitation needs due to COVID-19: update to May 31st, 2020. Eur J Phys Rehabil Med. 2020;56(4):508-514. doi:10.23736/S1973-9087.20.06435-7

RESPONSE: Thank you for this suggestion. As stated earlier, this study is strictly an epidemiological look at reinfection with no information on symptoms, treatments or interventions. This is not a paper on the burden of COVID nor do we have any data to support the role of therapies or modifications in the prevention of recurrence. However, we added your resources and the need for research on the relationship between lifestyle changes, like behavioral modifications and post-COVID therapies, and risk of recurrence in the Discussion section.

METHODS. The number of patients enrolled in a prospective study represents a study result. Please present these data in the Results section rather than the Methods Section.

RESPONSE: The only section of the Methods that describes number of patients is in the “Participants” sub-section in order to provide context, as this study’s sample was drawn from a larger prospective study. The number of patients enrolled in the larger prospective study is described in the Methods. This study’s number of patients is provided in the Results section, as you suggested and we agree that it represents a study result.

DISCUSSION. Please discuss the huge difference in percentage between females and males probably because women are much, much more likely to be in clinical roles in a healthcare setting – typically as nurses or medical assistants.

RESPONSE: Thank you for this suggestion. The Discussion section has been updated to reflect this.

RESULTS AND DISCUSSION. This section should be divided into two different sections. The manuscript form is not suitable for the journal without the suggested improvements.

RESPONSE: The Discussion section is, indeed, a separate sub-section, just nested within the Results and Discussion heading, per PLOS ONE’s style requirements as provided to us in this link in a prior round of reviews: https://journals.plos.org/plosone/s/file?id=wjVg/PLOSOne_formatting_sample_main_body.pdf

---

## [Editor Report · Decision Letter 2]

19 Dec 2021

Incidence of COVID-19 reinfection among Midwestern healthcare employees

PONE-D-21-26814R2

Dear Dr. Rivelli,

We’re pleased to inform you that your manuscript has been judged scientifically suitable for publication and will be formally accepted for publication once it meets all outstanding technical requirements.

Kind regards,

Alessandro de Sire, M.D.

Academic Editor

PLOS ONE

---

## [Editor Report · Acceptance letter]

23 Dec 2021

PONE-D-21-26814R2 

Incidence of COVID-19 reinfection among Midwestern healthcare employees 

Dear Dr. Rivelli:

I'm pleased to inform you that your manuscript has been deemed suitable for publication in PLOS ONE. Congratulations! Your manuscript is now with our production department. 

Kind regards, 

on behalf of

Prof. Alessandro de Sire 

Academic Editor

PLOS ONE